# DIRECT MOLECULAR CONFORMATION GENERATION

## ABSTRACT

Molecular conformation generation, which is to generate 3 dimensional coordinates of all the atoms in a molecule, is an important task for bioinformatics and pharmacology. Most existing machine learning based methods first predict interatomic distances and then generate conformations based on them. This two-stage approach has a potential limitation that the predicted distances may conflict with each other, e.g., violating the triangle inequality. In this work, we propose a method that directly outputs the coordinates of atoms, so that there is no violation of constraints. The conformation generator of our method stacks multiple blocks, and each block outputs a conformation which is then refined by the following block. We adopt the variational auto-encoder (VAE) framework and use a latent variable to generate diverse conformations. To handle the roto-translation equivariance, we adopt a loss that is invariant to rotation and translation of molecule coordinates, by computing the minimal achievable distance after any rotation and translation. Our method outperforms strong baselines on four public datasets, which shows the effectiveness of our method and the great potential of the direct approach. The code is released at `https://github.com/DirectMolecularConfGen/DMCG`.

## 1 INTRODUCTION

Molecular conformation generation aims to generate the 3D coordinates of all the atoms of molecules, which then can be used in molecular property prediction (Axelrod & Gomez-Bombarelli, 2021), docking (Roy et al., 2015), structure-based virtual screening (Kontoyianni, 2017), etc. Molecular conformation can be physically observed using X-ray crystallography, but it is prohibitively costly for industry-scale tasks (Mansimov et al., 2019). *Ab initio* methods, such as density functional theory (DFT) (Parr, 1980; Baseden & Tye, 2014), can accurately predict the molecular shapes, but take up to several hours per small molecule (Hu et al., 2021).To handle large-scale molecules, people turn to leverage the classical force fields, like UFF (Rappe et al., 1992) or MMFF (Halgren, 1996), to estimate conformations, which is efficient but unacceptably inaccurate (Kanal et al., 2018).

Recently, people start to explore machine learning methods to generate conformation. Simm & Hernández-Lobato (2020), Shi et al. (2020) and Shi et al. (2021) leveraged variational auto-encoder (VAE), flow-based models and score-based methods for conformation generation, respectively. With learned models, molecule conformations can be sampled independently (Simm & Hernández-Lobato, 2020; Shi et al., 2020) or using Langevin dynamics (Xu et al., 2021a; Shi et al., 2021). The common part of these methods is that they are all built upon the interatomic distances among atoms (i.e., the distance between atom pairs). Specifically, Simm & Hernández-Lobato (2020); Shi et al. (2020); Xu et al. (2021a;b) use various generative methods to model the distribution of interatomic distances, and then reconstruct conformations based on distances. Shi et al. (2021) leveraged a score-matching network to model the density gradient of interatomic distances. Moreover, Winter et al. (2021) use variational auto-encoder to first predict bond length, bond angle and dihedral angle and then reconstruct the coordinates based on the intermediate results. A major reason of using distance-based methods is that the interatomic distances are invariant to rotation and translation of the conformation. However, a potential drawback is that the predicted distances might conflict with each other. For example, they did not explicitly consider the triangle inequality for distances among any three atoms. The underlying degree-of-freedom of these distances is only $3N - 6$ ($N$ is the number of atoms in a molecule) while practically, these methods often generate distances with the degree-of-freedom of roughly $N(N - 1)/2$, which usually lead to violations of the triangle inequality. For example, we found that in GraphDG (Simm & Hernández-Lobato, 2020), a prevailing

representative of distance-based method, $8.65\%$ of molecular graphs in the GEOM-Drugs test set will produce distance matrix that violates the triangle inequality.

In this work, we explore the possibility to directly generate the coordinates of all atoms. This is a straightforward and more natural choice and avoids violations of the triangle inequality, and has demonstrated remarkable performance on protein structure prediction by the AlphaFold 2 (Jumper et al., 2021). We design a model that generates atom coordinates directly. The generator in our model stacks multiple blocks, and each block outputs a conformation which is then refined by the following block. A block consists of several modules that can encode the previous conformation as well as the features of bonds, atoms and global information. At the end of each block, we add a normalization layer that centers the coordinates at the origin. Since a molecule may have multiple conformations, we use the variational auto-encoder (VAE) framework which allows diverse generation. To realize roto-translation equivariance, i.e., rotating and translating a set of atom coordinates do not change the conformation, we adopt a loss that is invariant to rotation and translation of atom coordinates.

We conduct experiments on four benchmark settings, which are GEOM-QM9 and GEOM-Drugs with small-scale setting (Shi et al., 2021) and large-scale setting (Axelrod & Gomez-Bombarelli, 2021). Compared with previous methods, ours achieves state-of-the-art results on GEOM-Drugs (both small-scale and large-scale settings), and outperforms almost all previous baselines on GEOM-QM9, demonstrating the effectiveness of our method.

Our contributions are summarized as follows:

(1) We explore a new direction of conformation generation, that directly generates the coordinates of a molecular conformation without generating interatomic distances. We empirically show that directly generating conformation achieves state-of-the-art results on several tasks.

(2) We leverage a fine-grained loss function for training, that is invariant to the roto-translation of the conformation.

(3) We propose a new model that iteratively refines the conformations. Our model is inspired by multiple advanced architectures like GATv2 (Brody et al., 2021), graph network (GN) block (Battaglia et al., 2018), that can effectively model molecules.

## 2 PRELIMINARIES

In this section, we introduce the notations used in this work, give a formal definition of the molecular conformation generation problem, and briefly introduce how to align two conformations under rotation and translation.

*Notations*: Let $G = (V, E)$ denote a molecular graph, where $V$ and $E$ are collections of atoms and bonds, respectively. Specifically, $V = \{v_1, v_2, \cdots, v_{|V|}\}$ with the $i$-th atom $v_i$. Let $e_{ij}$ denote the bond between atom $v_i$ and $v_j$. Let $N(i)$ denote the neighbors of atom $i$, i.e., $N(i) = \{v_j \mid e_{ij} \in E\}$. We use $R$ to represent the conformation of $G$, where $R \in \mathbb{R}^{|V| \times 3}$. The $i$-th row of $R$ is the coordinate of atom $v_i$.

Let $\rho(\cdot)$ denote a roto-translation operation, i.e. an affine transformation $\rho(R) = R\boldsymbol{t} + \boldsymbol{b}$ where the orientation-preserving orthogonal transformation $\boldsymbol{t} \in \mathrm{SO}(3) \subset \mathbb{R}^{3 \times 3}$ represents a rotation and the vector $\boldsymbol{b} \in \mathbb{R}^{1 \times 3}$ represents a translation. Let $\mathrm{MLP}(\cdots)$ denote the multi-layer perception network, where the inputs are concatenated together and then processed by linear mapping, Batch Normalization (Ioffe & Szegedy, 2015) and ReLU activation sequentially.

*Problem definition*: Given a graph $G = (V, E)$, our task is to learn a mapping, that can output the coordinates $R$ of all atoms in $V$, i.e., $R \in \mathbb{R}^{|V| \times 3}$. [1]

*Matching loss*: Let $R_1$ and $R_2$ denote two conformations, both of which are $N \times 3$ matrices ($N$ is the number of rows in $R_1$ and $R_2$). The matching loss between $R_1$ and $R_2$ is defined as follows:

$$\ell_M(R_1, R_2) = \min_{\rho} \|\rho(R_1) - R_2\|_F^2, \tag{1}$$

---

[1] A molecule corresponds to different conformations at different energy level. To model such cases, the problem is to generate $R$ based on $G$ and a random noise $z$.

where $\|\cdot\|_F$ denotes the Frobenius norm, i.e., $\|A\|_F^2 = \sum_{i,j} |A_{ij}|^2$. The matching loss is invariant to the rotation and translation of either of the two input conformations. It is obviously nonnegative, and is zero if $R_1$ is obtained by a roto-translational operation of $R_2$.

Karney (2007) proposed to use quaternions to solve Eqn.(1). A quaternion $q$ is an extension of complex numbers, $q = q_0\mathbf{u} + q_1\mathbf{i} + q_2\mathbf{j} + q_3\mathbf{k}$, where $q_0, q_1, q_2, q_3$ are scalars and $\mathbf{u}, \mathbf{i}, \mathbf{j}, \mathbf{k}$ are orientation vectors. With quaternions, any rotation operation is specified by a $3 \times 3$ matrix, where each element in the matrix is the basic algebra of $q_0$ to $q_3$ and some constants. The solution to Eqn.(1) is the minimal eigenvalue of a $4 \times 4$ matrix obtained by the basic algebra on $R_1$ and $R_2$, the corresponding optimal rotation and translation $\rho^*$ can be obtained from the eigenvectors. Note that the best $\rho^*$ is related to both $R_1$ and $R_2$. To stabilize training, we set the gradients of $\rho^*$ w.r.t. the model parameter as zero. More details can be found in Karney (2007).

## 3 OUR METHOD

We introduce our proposed method in this section.

### 3.1 CONFORMATION GENERATION FRAMEWORK

Our goal can be formalized as learning to generate samples from the conditional distribution $p(R|G)$. We follow the variational auto-encoder (VAE) scheme (Kingma & Welling, 2014; Rezende et al., 2014; Sohn et al., 2015) as it enables flexible model architecture (vs. flow-based models), efficient (i.e., i.i.d) generation (vs. score-matching methods), and stable training and diverse sample generation (vs. adversarial methods). Given a molecule graph $G$, the conditional variant of VAE generates $R$ using a likelihood model $p(R|z, G)$ where $z$ is a latent code drawn from a prior $p(z) = \mathcal{N}(0, \mathbf{I})$. The corresponding $p(R|G) = \int p(z)p(R|z, G)\,\mathrm{d}z$ is however intractable to evaluate and maximize to fit data. VAE handles this by introducing a (conditional) inference model $q(z|R, G)$ which leads to the following identity:

$$\log p(R|G) = \mathbb{E}_{q(z|R,G)}\left[\log p(R|z, G)\right] - D_{\mathrm{KL}}(q(z|R, G)\|p(z)) + D_{\mathrm{KL}}(q(z|R, G)\|p(z|R, G)), \tag{2}$$

where $D_{\mathrm{KL}}$ denotes the Kullback-Leibler (KL) divergence, and $p(z|R, G) = \frac{p(z)p(R|z,G)}{\int p(z)p(R|z,G)\,\mathrm{d}z}$ is the true posterior. Due to the nonnegativity of the third term, the first two terms form a lower bound of $\log p(R|G)$. When $q(z|R, G)$ is properly chosen, the bound is tractable to optimize. Moreover, as the left-hand-side is independent of $q(z|R, G)$, tightening the bound (i.e., minimizing the third term) can be achieved by maximizing the bound w.r.t $q(z|R, G)$. Therefore, as a loss function to be minimized, the objective is:

$$\mathbb{E}_{q(z|R,G)}[-\log p(R|z, G)] + D_{\mathrm{KL}}(q(z|R, G)\|p(z)). \tag{3}$$

Now we develop our specification for the conformation generation task. We construct the likelihood model with regard to a decoded reference conformation $\hat{R}(z, G)$:

$$p(R|z, G) \propto \exp\left\{-\frac{1}{\gamma}\ell_M\big(\hat{R}(z, G), R\big)\right\}, \tag{4}$$

where $\gamma > 0$ is a variance parameter that we choose to fix as 1 (see Sec. 3.4 (1) for explanation). Note that the usage of $\ell_M$ guarantees all roto-translation equivalent conformations have the same probability. For the inference model, we choose

$$q(z|R, G) = \mathcal{N}(z|\mu_{R,G}, \Sigma_{R,G}), \tag{5}$$

where the conditional mean and variance are outputs from some encoder. It enables tractable loss optimization via reparameterization: $z \sim q(z|R, G)$ is equivalent to $z = \mu_{R,G} + \Sigma_{R,G}\epsilon$ where $\epsilon \sim \mathcal{N}(0, \mathbf{I})$. With the above specification, the loss function becomes:

$$\min \mathbb{E}_{\epsilon \sim \mathcal{N}(0,\mathbf{I})}\ell_M(\hat{R}(\mu_{R,G} + \Sigma_{R,G}\epsilon, G), R) + \beta D_{\mathrm{KL}}(\mathcal{N}(\mu_{R,G}, \Sigma_{R,G})\|\mathcal{N}(0, \mathbf{I})), \tag{6}$$

where the minimization is applied over all model parameters. Note that in Eqn.(6), an additional hyperparameter $\beta$ is introduced in the spirit of $\beta$-VAE (Higgins et al., 2016) to handle the strength of the prior regularization. The second term in Eqn.(6) has a closed-form expression for optimization.

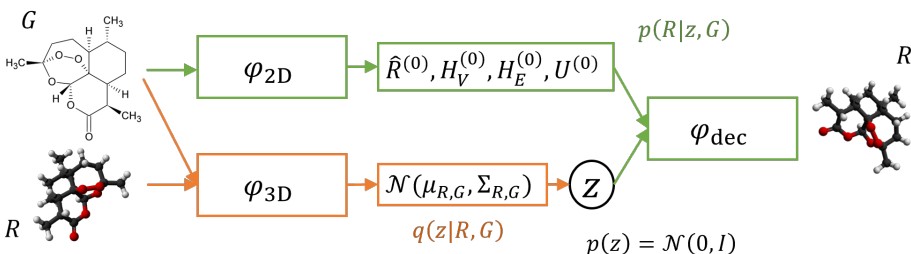

Figure 1: The workflow of our method. Green and orange lines represent $p(R|z, G)$ and $q(z|R, G)$ respectively.

## 3.2 WORKFLOW

According to Eqn.(3), we need a module $\varphi_{3D}$ to model $q(z|G, R)$, a module $\varphi_{dec}$ to model $p(R|z, G)$ and a module $\varphi_{2D}$ to model the conditional input $G$. In this subsection, we introduce the general workflow of how they work together (see Figure 1). Details of the $\varphi_{...}$ modules will be covered in the next subsection.

The training workflow is shown as follows:

(1) The encoder $\varphi_{2D}$ takes the molecular graph $G$ as its input, and outputs $d$-dimensional graph representations $H_V^{(0)} \in \mathbb{R}^{|V| \times d}$ for all atoms, $H_E^{(0)} \in \mathbb{R}^{|E| \times d}$ for all bonds, a global graph feature $U^{(0)} \in \mathbb{R}^d$, and initial conformation $\hat{R}^{(0)} \in \mathbb{R}^{|V| \times 3}$. Formally, $(H_V^{(0)}, H_E^{(0)}, U^{(0)}, \hat{R}^{(0)}) = \varphi_{2D}(G)$.

(2) We use another encoder $\varphi_{3D}$ to extract features of the conformation $R$ for constructing the conditional inference model $q(z|R, G)$. According to the above specification, this encoder only needs to output the mean and variance of the Gaussian, or formally, $(\mu_{R,G}, \Sigma_{R,G}) = \varphi_{3D}(R, G)$.

(3) We randomly sample a variable $z$ from the Gaussian distribution $\mathcal{N}(\mu_{R,G}, \Sigma_{R,G})$, and then feed $H_V^{(0)}, H_E^{(0)}, U^{(0)}, \hat{R}^{(0)}, z$ into the decoder $\varphi_{dec}$ to obtain the conformation $\hat{R}$. That is, $\hat{R}(z, G) = \varphi_{dec}(\varphi_{2d}(G), z) = \varphi_{dec}(H_V^{(0)}, H_E^{(0)}, \hat{R}^{(0)}, U^{(0)}, z)$. Note that sampling $z \sim \mathcal{N}(\mu_{R,G}, \Sigma_{R,G})$ is equivalent to sampling $\epsilon \sim \mathcal{N}(0, \mathbf{I})$ and then setting $z = \mu_{R,G} + \Sigma_{R,G}\epsilon$.

(4) After obtaining $\hat{R}(z, G)$ and and $\mathcal{N}(\mu_{R,G}, \Sigma_{R,G})$, we use Eqn.(6) as the objective function for training. Remind that $\hat{R}$ is related to $\varphi_{2D}, \varphi_{3D}, \varphi_{dec}$, and $\mu_{R,G}, \Sigma_{R,G}$ are related to $\varphi_{3D}$.

For the inference workflow, there are three steps in total: (1) Given a molecular graph $G$, we use $\varphi_{2D}$ to encode $G$ and obtain $\hat{R}^{(0)}, H_V^{(0)}, H_E^{(0)}, U^{(0)}$; (2) sample a random variable $z$ from Gaussian $\mathcal{N}(0, \mathbf{I})$; (3) feed $\hat{R}^{(0)}, H_V^{(0)}, H_E^{(0)}, U^{(0)}, z$ into $\varphi_{dec}$ and obtain the eventual conformation $\hat{R}(z, G)$. Note that $\varphi_{3D}$ is not used in inference phase.

## 3.3 DETAILED MODEL ARCHITECTURE

The encoders $\varphi_{2D}, \varphi_{3D}$ and the decoder $\varphi_{dec}$ share the same architecture. They are all stacks of $L$ identical blocks. We introduce the $l$-th block in the decoder $\varphi_{dec}$, and leave the introduction of $\varphi_{2D}$ and $\varphi_{3D}$ in Appendix A. In contrast to previous work, we introduce some novel designs that handle the task at a finer level. (1) The conformation is iteratively refined by each decoding block. Specifically, we use a normalization technique that moves the center of each decoded conformation to the origin. (2) We use more advanced model components, like attention models (Brody et al., 2021) and the GN block (Battaglia et al., 2018), that can more effectively represent the molecules.

The detailed architecture is shown in Figure 2. The inputs include the following information output by the $(l-1)$-th block: the conformation $\hat{R}^{(l-1)}$, atom features $H_V^{(l-1)} \in \mathbb{R}^{|V| \times 3}$, edge features $H_E^{(l-1)}$ and the global feature $U^{(l-1)}$. Note $H_V^{(0)}, H_E^{(0)}, U^{(0)}$ and $\hat{R}^{(0)}$ are the outputs of $\varphi_{2D}$. $\hat{R}^{(l-1)}$ is first mapped by MLP and get $F^{(l)} = \text{MLP}(\hat{R}^{(l-1)})$. The atom features are then fused with the coordinate features by $H_V^{(l-1)} \leftarrow H_V^{(l-1)} + F^{(l)} + z$, where $z \sim \mathcal{N}(\mu_{R,G}, \Sigma_{R,G})$.

We use a variant of the GN block (Battaglia et al., 2018) as the backbone of the model due to its superior performance in molecular modeling. In each block, we update the bond features, atom fea-

Figure 2: Network architecture of the $l$-th block. Yellow dashed boxes are inputs, orange boxes denote the operations, and green boxes are outputs.

tures and global features sequentially. Each block will output its own prediction on the coordinates of atoms. For ease of reference, we use $h_i^{(l)}$ (single subscript) to denote the feature of atom $i$ output by the $l$-th block, and use $h_{ij}^{(l)}$ (two subscripts) to denote the bond features between atom $i$ and $j$ output by the $l$-th block. The workflow of the $l$-th block is shown as follows:

(1) *Update bond features*: For each bond feature $h_{ij}^{(l-1)}$ in $H_E^{(l-1)}$, it is updated by

$$h_{ij}^{(l)} = h_{ij}^{(l-1)} + \texttt{MLP}(h_i^{(l-1)}, h_j^{(l-1)}, h_{ij}^{(l-1)}, U^{(l-1)}). \tag{7}$$

(2) *Update atom features*: The atom features are updated using an attentive way. For any $i \in [|V|]$,

$$\bar{h}_i^{(l)} = \sum_{j \in \mathcal{N}(i)} \alpha_j W_v \texttt{concat}(h_{ij}^{(l)}, h_j^{(l-1)});$$

$$\alpha_j \propto \exp(\boldsymbol{a}^\top \texttt{LeakyReLU}(W_q h_i^{(l-1)} + W_k \texttt{concat}(h_j^{(l-1)}, h_{ij}^l))); \tag{8}$$

$$h_i^{(l)} = h_i^{(l-1)} + \texttt{MLP}\left(h_i^{(l-1)}, \bar{h}_i^{(l)}, U^{(l-1)}\right).$$

In Eqn.(8), $\boldsymbol{a}$, $W_q$, $W_v$ and $W_k$ are the parameters to be learned and $\texttt{concat}(\cdot, \cdot)$ is the concatenation of two vectors. For atom $v_i$, we first use GATv2 (Brody et al., 2021) to aggregate the features from its connected bonds and obtain $\bar{h}_i$, and then update $v_i$ based on $\bar{h}_i^{(l)}$, $h_i^{(l-1)}$ and $U^{(l-1)}$.

(3) *Update global features*: The global feature is updated as follows:

$$U^{(l)} = U^{(l-1)} + \texttt{MLP}\left(\frac{1}{|V|}\sum_{i=1}^{|V|} h_i^{(l)}, \frac{1}{|E|}\sum_{i,j} h_{ij}^{(l)}, U^{(l-1)}\right). \tag{9}$$

(4) *Output the prediction*: After obtaining the new features of the $l$-th block, for any atom $i \in [|V|]$, it predicts the conformations $\hat{R}_i^{(l)}$ as follows:

$$\bar{R}_i^{(l)} = \texttt{MLP}(h_i^{(l)}), \quad m^{(l)} = \frac{1}{|V|}\sum_{j=1}^{|V|} \bar{R}_j^{(l)}, \quad \hat{R}_i^{(l)} = \bar{R}_i^{(l)} - m^{(l)} + \hat{R}_i^{(l-1)}. \tag{10}$$

An important step in Eqn.(10) is that, after making initial prediction $\bar{R}_i^{(l)}$, we calculate its center and normalize their coordinates by moving the center to the origin. We add this kind of normalization to ensure that input coordinates of each block are in reasonable numeric ranges.

We use $\hat{R}^{(L)}$ output by the last block in $\varphi_{\text{dec}}$ as the final prediction of the conformation.

### 3.4 DISCUSSIONS

In this section, we discuss the relation between our method and two previous work.

*Comparison with CVGAE*: Mansimov et al. (2019) developed an early attempt to generate conformation by raw coordinates, but its performance is not as desired as distance-based methods developed afterwards (Shi et al., 2020; Simm & Hernández-Lobato, 2020). Our method, pursuing the same spirit, makes several finer designs and implementations.

(1) Under the VAE formulation, Mansimov et al. (2019) also leaned the variance parameter $\gamma$ of the likelihood model. i.e., the $\gamma$ in Eqn.(4). However, a decent VAE analysis (Dai & Wipf, 2019,

Thm. 3) revealed that the VAE objective prefers a vanishing $\gamma$. This is desired in the ideal case when the encoder and decoder are optimized perfectly (Dai & Wipf, 2019, Thm. 4), but in practice it would distract the optimizer towards a "lazy way" to focus on $\gamma$ and stagnate the optimization of the encoder and decoder. Therefore, we choose to fix $\gamma$. When $\gamma$ is fixed, the loss function is equivalent to using $\gamma = 1$ and a $\beta$ parameter as the formulation in Eqn.(6). We hence choose a small value of $\beta$ corresponding to fixing $\gamma$ to a small value in the original formulation (i.e. without $\beta$) as advocated (Dai & Wipf, 2019).

(2) Our model consists of more advanced modules, including GATv2 (Brody et al., 2021) and GN block (Battaglia et al., 2018) to better model the input. In comparison, Mansimov et al. (2019) mainly leveraged GRU (Bahdanau et al., 2015) and its variant on graphs, which is outperformed by the modules used in our model.

(3) We iteratively refine the output of each block, while (Mansimov et al., 2019) only outputs the conformation in the last layer.

*Comparison with ConfGF*: Shi et al. (2021) used score matching to model the gradients w.r.t interatomic distances, and then recover the coordinates based on the gradients. That is, (Shi et al., 2021) still tries to model the interatomic distances, while we completely abandon modeling the distances. We propose a new direction for molecular conformation generation.

# 4 EXPERIMENT

## 4.1 SETTINGS

*Datasets*: Following prior works (Xu et al., 2021a; Shi et al., 2021), we use the GEOM-QM9 and GEOM-Drugs datasets (Axelrod & Gomez-Bombarelli, 2021) for conformation generation. We verify our method on both small-scale setting and large-scale setting. First, we use the same datasets provided by Shi et al. (2021) for fair comparison with prior works. The training, validation and test sets of the two datasets consist of 200K, 2.5K and 22408 (for GEOM-QM9)/14324 (for GEOM-Drugs) molecule-conformation pairs respectively. After that, we sample larger datasets from the original GEOM to validate the scalability of our method. We use all data in GEOM-QM9 and $2.2M$ molecule-conformation pairs for GEOM-Drugs. The numbers of training, validation and test sets for the larger GEOM-QM9 setting are 1.37M, 165K and 174K, and those for larger GEOM-Drugs are 2M, 100K and 100K.

*Model configuration*: All of $\varphi_{2D}$, $\varphi_{3D}$ and $\varphi_{dec}$ have the same number of blocks. The dimension $d$ of the features is 256. Inspired by the feed-forward layer in Transformer (Vaswani et al., 2017), MLP also consists of two sub-layers, where the first one maps the input features from dimension 256 to hidden states, followed by Batch Normalization and ReLU activation. Then the hidden states is mapped to 256 again using linear mapping. More parameters are summarized in Appendix B.

*Evaluation*: Assume in the test set, the molecule $x$ has $N_x$ conformations. Following Shi et al. (2020; 2021), for each molecule $x$ in the test set, we generate $2N_x$ conformations. Let $\mathbb{S}_g(\mathcal{G})$ and $\mathbb{S}_r(\mathcal{G})$ denote all generated and groundtruth conformations respectively. We use coverage score (COV) and matching score (MAT) to evaluate the generation quality. Mathematically,

$$\text{COV}(\mathbb{S}_g(\mathcal{G}), \mathbb{S}_r(\mathcal{G})) = \frac{1}{|\mathbb{S}_r|} \left| \{ \boldsymbol{R} \in \mathbb{S}_r \mid \ell_M (\boldsymbol{R}, \boldsymbol{R}') < \delta, \exists \boldsymbol{R}' \in \mathbb{S}_g \} \right|; \tag{11}$$

$$\text{MAT}(\mathbb{S}_g(\mathcal{G}), \mathbb{S}_r(\mathcal{G})) = \frac{1}{|\mathbb{S}_r|} \sum_{\boldsymbol{R}' \in \mathbb{S}_r} \min_{\boldsymbol{R} \in \mathbb{S}_g} \ell_M (\boldsymbol{R}, \boldsymbol{R}'), \tag{12}$$

where $\ell_M$ is defined in Eqn.(1). A good method should have a high COV score and a low MAT score. Following (Shi et al., 2021), the $\delta$'s are set as $0.5$ and $1.25$ for QM9 and Drugs, respectively.

*Baselines*: (1) RDKit, which is a widely used toolkit and generates the conformation based on the force fields; (2) CVGAE (Mansimov et al., 2019), which is an early attempt to generate raw coordinates; (3) GraphDG (Simm & Hernández-Lobato, 2020), a representative distance-based method with VAE; (4) CGCF (Xu et al., 2021a), which is another distance-based method leveraging continuous normalizing flow; (5) ConfVAE (Xu et al., 2021b), an end-to-end framework for molecular conformation generation, which still uses the pairwise distances among atoms as intermediate variables;

(6) ConfGF (Shi et al., 2021), which uses score matching to generate the gradients w.r.t. distances and then recover the conformation. Note that ConfGF leverages Langevin dynamics, which requires much more additional decoding time.

## 4.2 Results

The results on the small-scale and large-scale datasets are in Table 1 and Table 2 respectively.

| Dataset | QM9 | | | | Drugs | | | |
|---|---|---|---|---|---|---|---|---|
| | COV(%)↑ | | MAT (Å)↓ | | COV(%)↑ | | MAT (Å)↓ | |
| Methods | Mean | Median | Mean | Median | Mean | Median | Mean | Median |
| RDkit | 83.26 | 90.78 | 0.3447 | 0.2935 | 60.91 | 65.70 | 1.2026 | 1.1252 |
| CVGAE | 0.09 | 0.00 | 1.6713 | 1.6088 | 0.00 | 0.00 | 3.0702 | 2.9937 |
| GraphDG | 73.33 | 84.21 | 0.4245 | 0.3973 | 8.27 | 0.00 | 1.9722 | 1.9845 |
| CGCF | 78.05 | 82.48 | 0.4219 | 0.3900 | 53.96 | 57.06 | 1.2487 | 1.2247 |
| ConfVAE | 80.42 | 85.31 | 0.4066 | 0.3891 | 53.14 | 53.98 | 1.2392 | 1.2447 |
| ConfGF | **88.49** | 94.13 | **0.2673** | 0.2685 | 62.15 | 70.93 | 1.1629 | 1.1596 |
| Ours | 81.44 | **96.61** | 0.2879 | **0.2428** | **77.78** | **86.09** | **1.0657** | **1.0563** |

Table 1: Experimental results on small-scale datasets.

| Dataset | QM9 | | | | Drugs | | | |
|---|---|---|---|---|---|---|---|---|
| | COV(%)↑ | | MAT (Å)↓ | | COV(%)↑ | | MAT (Å)↓ | |
| Methods | Mean | Median | Mean | Median | Mean | Median | Mean | Median |
| RDkit | 81.61 | 85.71 | 0.2643 | 0.2472 | 69.42 | 77.45 | 1.0880 | 1.0333 |
| CVGAE | 0.00 | 0.00 | 1.4687 | 1.3758 | 0.00 | 0.00 | 2.6501 | 2.5969 |
| GraphDG | 13.48 | 5.71 | 0.9511 | 0.9180 | 1.95 | 0.00 | 2.6133 | 2.6132 |
| CGCF | 81.48 | 86.95 | 0.3598 | 0.3684 | 57.47 | 62.09 | 1.2205 | 1.2003 |
| ConfVAE | 80.18 | 85.87 | 0.3684 | 0.3776 | 57.63 | 63.75 | 1.2125 | 1.1986 |
| ConfGF | **89.21** | 95.12 | 0.2809 | 0.2837 | 70.92 | 85.71 | 1.0940 | 1.0917 |
| Ours | 86.58 | **100.00** | **0.2457** | **0.1756** | **90.68** | **100.00** | **0.8934** | **0.8703** |

Table 2: Experimental results on large-scale datasets.

We have the following observations:

(1) On the four settings in Table 1 and Table 2, our method achieves state-of-the-art results on most of them, i.e., the two large-scale settings and small dataset setting on GEOM-drug. The only exception is that our method is not so good as ConfGF in terms of the COV mean and MAT mean on small-scale QM9, and the mean COV score on large-scale QM9. The median COV(%) being 100% means that for more than half of the reference molecules, these exist generated molecules that are close to them within a predefined threshold. These results already showed the effectiveness and scalability of our method.

(2) Our method achieves more improvement on molecules with more heavy atoms. Take the results in Table 1 as an example. On average, QM9 and Drugs have 8.8 and 24.9 heavy atoms respectively. In terms of MAT median, on QM9, our method improves ConfGF by 0.026 points, while on Drugs, the improvement is 0.103. To verify our conjecture, on Drugs, we categorize the molecules based on their numbers of heavy atoms. The number of heavy atoms in the $i$-th group lie in $[10i+1, 10(i+1)]$. We compare our method against ConfGF and GraphDG. The results are in Table 3.

| Metric | COV(%)↑ | | | | MAT(Å)↓ | | | |
|---|---|---|---|---|---|---|---|---|
| | $i=1$ | $i=2$ | $i=3$ | average | $i=1$ | $i=2$ | $i=3$ | average |
| ConfGf | 99.95 | 66.28 | 15.34 | 62.54 | 0.7764 | 1.1510 | 1.5345 | 1.1637 |
| GraphDG | 15.11 | 1.78 | 0.0 | 3.12 | 2.0578 | 2.5863 | 2.9849 | 2.5847 |
| Ours | 98.13 | 80.18 | 51.72 | 77.78 | 0.8502 | 1.0552 | 1.2737 | 1.0657 |

Table 3: COV and MAT mean scores w.r.t. numbers of heavy atoms on small-scale GEOM-Drugs. The $i$ indicates that the heavy atom number lies in range $[10i + 1, 10(i + 1)]$, $i \in \{1, 2, 3\}$.

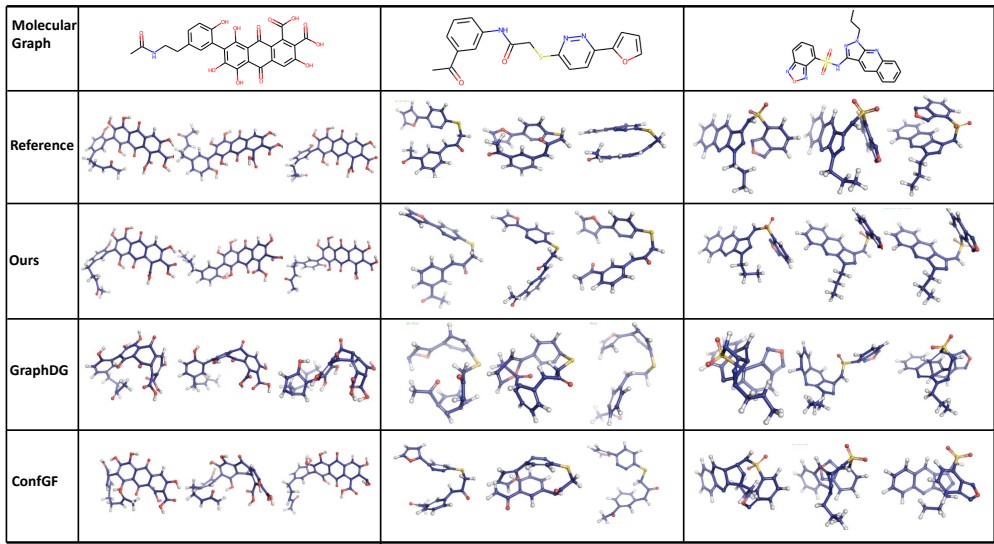

Figure 3: Visualization of different conformations.

We can observe that among the three algorithms, ConfGF achieves the best results on the molecules whose numbers of heavy atoms lie in $[11, 20]$, and our method is slightly worse than it. However, when the numbers of heavy atoms are larger than 21, our method significantly outperforms ConfGF and GraphDG. This is consistent with our conjecture, and demonstrates the potential of directly generating coordinates for molecules with more heavy atoms. More discussions are in Appendix C.5.

(3) Our method is much more sample efficient than ConfGF. In (Shi et al., 2021), generating a conformation requires 5000 sequential forward steps. With our method, we only need to sample one variable from $\mathcal{N}(0, \boldsymbol{I})$ and then generate a corresponding conformation by forwarding once. For a fair comparison, following the official implementation of ConfGF, we split the test sets of small-scale GEOM-QM9 and GEOM-Drugs into 200 batches. ConfGF requires 8511.60 and 11830.42 seconds to decode QM9 and Drugs test sets, while our method only requires 32.68 and 54.89 seconds respectively. That is, our method speeds up the decoding more than 200 times.

In Figure 3, we visualize the conformation of different methods. We randomly select four molecules from the small-scale GEOM-drug dataset, generate several conformations, and visualize the best-aligned ones with the groundtruth. We can see that our method can generate high-quality conformations than previous methods, which are the most similar to the groundtruth.

## 4.3 PROPERTY PREDICTION

We conduct experiments on the property prediction task, which is to predict molecular properties based on an ensemble of generated conformations (Axelrod & Gomez-Bombarelli, 2021). We first randomly choose 30 molecules from QM9 and Drugs test sets, and then sample 50 conformations for each molecule using RDkit, ConfGF and our method as initial coordinates. Next, we use the package Psi4 (Smith et al., 2020) to calculate the energy, HOMO and LUMO for each generated conformation and groundtruth conformation. After that, we calculate ensemble properties including average energy $\bar{E}$, lowest energy $E_{\min}$, average HOMO-LUMO gap $\overline{\Delta\epsilon}$, minimum gap $\Delta\epsilon_{\min}$, and maximum gap $\Delta\epsilon_{\max}$ for the conformations. Finally, we use the mean absolute error to measure the gap of the above properties between the groundtruth and generated conformations. Although a better choice is to use Boltzmann distribution to re-weight each conformation, due to the lack of such the distribution in the dataset, we are not able to use it now and use the average value instead.

| Methods | QM9 | | | | | Drugs | | | | |
|---|---|---|---|---|---|---|---|---|---|---|
| | $\bar{E}$ | $E_{\min}$ | $\overline{\Delta\epsilon}$ | $\Delta\epsilon_{\min}$ | $\Delta\epsilon_{\max}$ | $\bar{E}$ | $E_{min}$ | $\overline{\Delta\epsilon}$ | $\Delta\epsilon_{\min}$ | $\Delta\epsilon_{\max}$ |
| Rdkit | 0.0902 | 0.0534 | 0.2001 | 0.2215 | 0.1837 | 3.4808 | 0.1669 | 0.4082 | 2.8640 | 0.2720 |
| GraphDG | 0.1108 | 0.0789 | 0.2309 | 0.3886 | 0.2431 | 8.9561 | 0.4485 | 0.9217 | 3.2755 | 0.4519 |
| ConfGF | 0.0719 | 0.0295 | 0.1533 | 0.2302 | 0.1640 | 2.9843 | 0.1763 | 0.2188 | 2.6481 | 0.2422 |
| Ours | 0.0656 | 0.0258 | 0.1358 | 0.2156 | 0.1638 | 0.1796 | 0.1468 | 0.2073 | 0.6740 | 0.2243 |

Table 4: Mean absolute error of predicted ensemble properties. Unit: eV.

| Methods | COV(%)↑ | | MAT (Å)↓ | |
| --- | --- | --- | --- | --- |
| | Mean | Median | Mean | Median |
| Ours | **77.78** | **86.09** | **1.0657** | **1.0563** |
| Ours w/o attention | 72.18 | 82.76 | 1.1257 | 1.1138 |
| Ours w/o conformation normalization | 61.82 | 64.76 | 1.2045 | 1.1974 |
| Ours w/ FF | 85.00 | 92.05 | 0.8959 | 0.8992 |

Table 5: Ablation study on small-scale GEOM-Drugs.

The results are shown in Table 4. On QM9, our method achieves slightly better results than ConfGF, while on Drugs, the results are significantly better than ConfGF and GraphDG. This shows that our method can provide good conformations for property prediction tasks.

## 4.4 ABLATION STUDY

To further verify our algorithm, we conduct the following ablation study: (1) We remove the attentive node aggregation; instead, it is replaced by a simple `MLP` network. That is, Eqn.(8) is replaced by

$$h_i^{(l)} = h_i^{(l-1)} + \texttt{MLP}(h_i^{(l-1)}, U^{(l-1)}, \frac{1}{|\mathcal{N}(i)|} \sum_{j \in \mathcal{N}(i)} h_j^{(l-1)}); \quad (13)$$

(2) We remove the normalization step in Eqn.(10), i.e., the $m^{(l)}$ is not used; (3) We further use the force field in RDKit to refine the output of our method. The results are summarized in Table 5. We can see that: (1) Without attentively aggregating the atom features, the performances drops: The mean COV drops $5.6$ points and MAT score increases $0.06$ points. (2) Without the conformation normalization, the performance is greatly effected, which shows the importance of this step; (3) Our method can still be refined by the RDKit, which shows that our method can be combined with the classic methods for more improvements.

Finally, we compute the COV and MAT scores of $\hat{R}^{(l)}$ against the groundtruth, which is the output conformation of the $l$-th block in the decoder. $\hat{R}^{(0)}$ is the output of $\varphi_{2D}$. The results are shown in Figure 4. We can see that iteratively refining the conformations can improve the performances, which shows the effectiveness of our design.

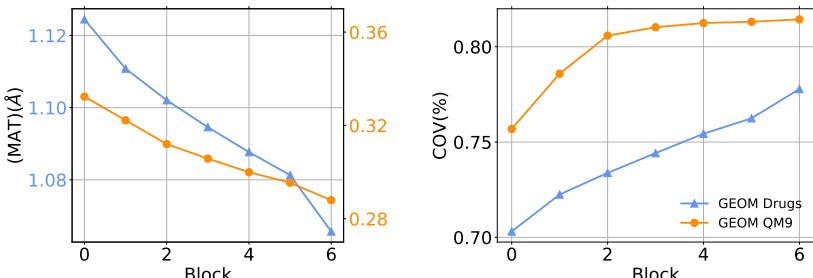

Figure 4: MAT and COV scores of $\hat{R}^{(l)}$ against the grondtruth.

## 5 CONCLUSIONS AND FUTURE WORK

In this work, we propose a new method, that directly generates the coordinates of conformations. For this purpose, we leverage a loss function that is invariant to roto-translation, and design a new model with many advanced modules (i.e., GATv2, GN block) that can iteratively refine the conformations. Experimental results on both small-scale and large-scale GEOM-QM9 and GEOM-Drugs demonstrate the effectiveness of our method.

For future work, there are many interesting directions. First, we will combine with more generative methods like flow-based model, Langevin dynamics and so on. Second, during case analyzing, we found that there are still some cases that current methods cannot successfully handle (e.g., rotatable rings) and we will improve them in the future (see Appendix C.3). Third, current methods are mainly non-autoregressive, where all coordinates are generated simultaneously. We will study the autoregressive setting so as to further improve the accuracy. Fourth, the current loss function is not invariant to permutations of symmetric atoms, and we will improve it in the future.

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

## A  DETAILS OF $\varphi_{2D}$ AND $\varphi_{3D}$

The model architectures of $\varphi_{2D}$ and $\varphi_{3D}$ are similar to $\varphi_{dec}$, with the following differences.

Comparing $\varphi_{2D}$ with $\varphi_{dec}$, the differences are the initial conformation $\hat{R}^{(0)}$ and initial features (i.e., the $H_V^{(0)}$, $H_E^{(0)}$ and $U^{(0)}$). $\varphi_{2D}$ takes a random conformation sampled from uniform distribution in $[-1, 1]$ as input. The initial atom and edge features are the embeddings of the atoms and edges respectively. $\varphi_{2D}$ will also output a prediction of the conformation. Note that the random variable $z$ sampled from Gaussian $\mathcal{N}(\mu_{R,G}, \Sigma_{R,G})$ is not used in $\varphi_{2D}$.

Comparing $\varphi_{3D}$ with $\varphi_{dec}$, the differences are the initial conformation $\hat{R}^{(0)}$, initial features (i.e., the $H_V^{(0)}$, $H_E^{(0)}$ and $U^{(0)}$) too. $\varphi_{3D}$ takes the groundtruth conformation as input. The initial atom and edge features are the embeddings of the atoms and edges respectively. Another difference is that the fourth step of $\varphi_{dec}$, i.e., Eqn.(10), is not used. More details are available in our code.

## B  MORE EXPERIMENT DETAILS

We use AdamW optimizer (Loshchilov & Hutter, 2019) with initial learning rate $\eta_0 = 2 \times 10^{-4}$ and weight decay 0.01. In the first 4000 iterations, the learning rate is linearly increased from $10^{-6}$ to $2 \times 10^{-4}$. After that, we use cosine learning rate scheduler (Loshchilov & Hutter, 2016), where the learning rate at the $t$-th iteration is $\eta_0(1 + \cos(\pi \frac{t}{T}))/2$, where $T$ is the half of the period (i.e., the iteration numbers of 10 epochs in our setting). Similarly, we also use the cosine scheduler to dynamically set the $\beta$ at range $[0.0001, 0.008]$. The batch size is fixed as 128. All models are trained for 100 epochs. For the two small-scale settings, the experiments are conducted on a single V100 GPU. For the two large-scale settings, we use two V100 GPUs for experiments. The detailed hyper-parameters are described in Table 6.

| | Small-Scale | Large-Scale |
|---|---|---|
| Layer number | 3 | 6 |
| Dropout | 0.1 | 0.1 |
| Learning rate | 2e-4 | 2e-4 |
| Batch size | 128 | 128 |
| Epoch | 100 | 100 |
| Vae $\beta$ Min | 0.0001 | 0.001 |
| Vae $\beta$ Max | $\{0.001, 0.002, 0.004, 0.008, 0.01\}$ | $\{0.005, 0.01, 0.02, 0.04, 0.05\}$ |
| Latent size | 256 | 256 |
| Hidden dimension | 512 | 1024 |
| GPU number | $1\times$ NVIDIA V100 | $2\times$ NVIDIA V100 |

Table 6: Hyper-parameters for our experiments

## C  MORE EXPERIMENTAL RESULTS

### C.1  COMBINATION WITH DISTANCE-BASED AND ANGLE-BASED LOSS FUNCTIONS

In addition to the matching loss defined in Eqn.(1) which is related to coordinates only, one may be curious about whether using distance-based loss and angle-based can further improve the performance, since the latter two are equivariant to the transformation of coordinates. For ease of

reference, let $R_i$ denote the groundtruth coordinate of atom $v_i$ and $\hat{R}_i$ denote the predicted coordinate of atom $v_i$. Remind in Section 2, we use $E$ to denote the collection of all bonds. We also define $E_2$ as $\{(i,j,k)|(i,j) \in E, (i,k) \in E, k \neq j\}$.

Following (Winter et al., 2021) we use the following two functions:

$$\ell_{\text{angle}} = \frac{1}{|E_2|} \sum_{(i,j,k) \in E_2,} \| \text{cosine}(R_j - R_i, R_k - R_i) - \text{cosine}(\hat{R}_j - \hat{R}_i, \hat{R}_k - \hat{R}_i) \|_F^2, \quad (14)$$

$$\ell_{\text{bond}} = \frac{1}{|E|} \sum_{(i,j) \in E} \left( \text{distance}(R_j, R_i) - \text{distance}(\hat{R}_j - \hat{R}_i) \right)^2, \quad (15)$$

where $\text{cosine}(a,b) = \frac{a^\top b}{\|a\|\|b\|}$ and $\text{distance}(a,b) = \|a - b\|$, $a$ and $b$ are two vectors. That is, we apply additional constraints to bond length and bond angles. Please note that with the above two auxiliary loss functions, our method still generates coordinates directly and does not need to generate intermediate distances and angles.

We set the weight of the above two loss functions as $0.1$, and add them to Eqn.(6). We conduct experiments on small-scale QM9 and Drugs. The results are reported in Table 7, which are in the row starting with "Ours + Aux". We can see after using those two auxiliary loss functions which are invariant to the roto-translational operations on the coordinates, the results can be further improved. We will further explore this direction in the future.

| Dataset | QM9 | | | | Drugs | | | |
|---|---|---|---|---|---|---|---|---|
| | COV(%)↑ | | MAT (Å)↓ | | COV(%)↑ | | MAT (Å)↓ | |
| Methods | Mean | Median | Mean | Median | Mean | Median | Mean | Median |
| RDkit | 83.26 | 90.78 | 0.3447 | 0.2935 | 60.91 | 65.70 | 1.2026 | 1.1252 |
| CVGAE | 0.09 | 0.00 | 1.6713 | 1.6088 | 0.00 | 0.00 | 3.0702 | 2.9937 |
| GraphDG | 73.33 | 84.21 | 0.4245 | 0.3973 | 8.27 | 0.00 | 1.9722 | 1.9845 |
| CGCF | 78.05 | 82.48 | 0.4219 | 0.3900 | 53.96 | 57.06 | 1.2487 | 1.2247 |
| ConfVAE | 80.42 | 85.31 | 0.4066 | 0.3891 | 53.14 | 53.98 | 1.2392 | 1.2447 |
| ConfGF | 88.49 | 94.13 | 0.2673 | 0.2685 | 62.15 | 70.93 | 1.1629 | 1.1596 |
| Ours | 81.44 | 96.61 | 0.2879 | 0.2428 | 77.78 | 86.09 | 1.0657 | 1.0563 |
| Ours + Aux | **91.97** | **98.66** | **0.2388** | **0.2231** | **92.45** | **98.70** | **0.8983** | **0.9016** |

Table 7: Combination with distance-based and angle-based loss functions.

## C.2 PROPERTY PREDICTION

Following Shi et al. (2021), besides energy, we also calculate HOMO and LUMO of all the conformation using Psi4. Note that all the generated conformations are refined by MMFF. Then we will calculate ensemble property average energy $\overline{E}$, lowest energy $E_{min}$, average HOMO-LUMO gap $\overline{\Delta\epsilon}$, minimum gap $\Delta\epsilon_{min}$ and maximum gap $\Delta\epsilon_{max}$ for each method and use mean absolute error (MAE) and median absolute error with ground-truth conformations to measure the accuracy. The results are showed in Table 8 and Table 9. As the results shown, our method achieves the best accuracy in general.

| Methods | QM9 | | | | | Drugs | | | | |
|---|---|---|---|---|---|---|---|---|---|---|
| | $\overline{E}$ | $E_{min}$ | $\overline{\Delta\epsilon}$ | $\Delta\epsilon_{min}$ | $\Delta\epsilon_{max}$ | $\overline{E}$ | $E_{min}$ | $\overline{\Delta\epsilon}$ | $\Delta\epsilon_{min}$ | $\Delta\epsilon_{max}$ |
| Rdkit | 0.0507 | 0.0276 | 0.0853 | 0.1724 | 0.1384 | 0.1720 | 0.1255 | 0.1872 | 0.1790 | 0.2188 |
| GraphDG | 0.0859 | 0.0567 | 0.1723 | 0.2569 | 0.1977 | 5.6440 | 0.3520 | 0.6502 | 2.9129 | 0.2677 |
| ConfGF | 0.0458 | 0.0229 | 0.1198 | 0.1798 | 0.1531 | 2.7344 | 0.1379 | 0.3171 | 2.8166 | 0.1820 |
| Ours | 0.0418 | 0.0213 | 0.1163 | 0.1681 | 0.1450 | 0.1573 | 0.1129 | 0.1724 | 0.5469 | 0.1780 |

Table 8: Mean absolute error of predicted ensemble properties. Unit: eV.

| Methods | QM9 | | | | | Drugs | | | | |
|---|---|---|---|---|---|---|---|---|---|---|
| | $\overline{E}$ | $E_{min}$ | $\overline{\Delta\epsilon}$ | $\Delta\epsilon_{min}$ | $\Delta\epsilon_{max}$ | $\overline{E}$ | $E_{min}$ | $\overline{\Delta\epsilon}$ | $\Delta\epsilon_{min}$ | $\Delta\epsilon_{max}$ |
| Rdkit | 0.0325 | 0.0233 | 0.0814 | 0.0917 | 0.1103 | 0.1619 | 0.0879 | 0.1418 | 0.1432 | 0.1179 |
| GraphDG | 0.0624 | 0.0309 | 0.1430 | 0.1562 | 0.1704 | 4.1807 | 0.2190 | 0.4412 | 3.3542 | 0.2127 |
| ConfGF | 0.0348 | 0.0209 | 0.0991 | 0.1497 | 0.1534 | 2.7917 | 0.0725 | 0.2700 | 2.9545 | 0.1972 |
| Ours | 0.0318 | 0.0160 | 0.1174 | 0.1336 | 0.1501 | 0.0695 | 0.0618 | 0.1431 | 0.1425 | 0.1043 |

Table 9: Median absolute error of predicted ensemble properties. Unit: eV.

## C.3 FAILURE CASES

We notice some failure cases in our model prediction. Two examples are shown in Figure 5. We can see that there exists a rotatable ring at the end of a molecule, where the ring is symmetric to the bond connecting itself to the rest of the molecule. Our method fails to generate the coordinates of such rings, but simply puts them in a line. This is because the model is trapped into local optimal. By using the additional loss functions as we introduced Appendix C.1, we can successfully recover the conformations of those rings (see the row Ours + Aux). We will keep exploring along this direction.

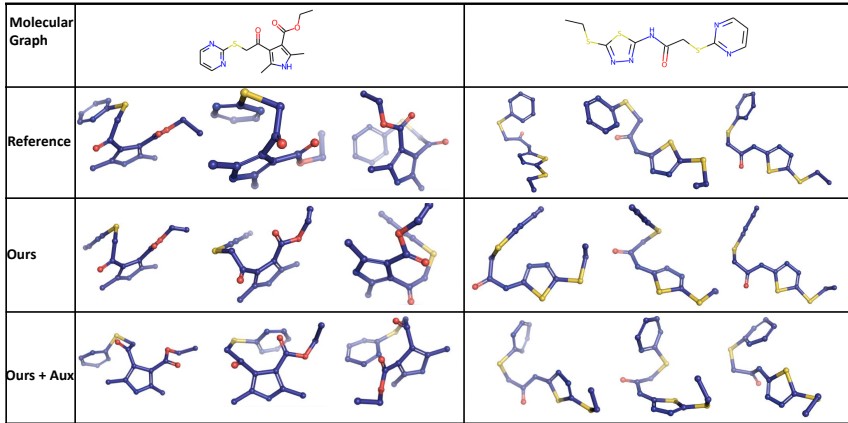

Figure 5: Failure case of our method. "Our + Aux" means that we add two auxiliary loss to our method, and these two losses improve our method effectively.

## C.4 STUDY OF MODEL PARAMETERS

In this section, we compare the performances of our method and ConfGF. By default, our model has 13.29M parameters, and the ConfGF model has 0.81M parameters. We explore the following two settings: (1) we reduce our model size to 0.98M by reducing hidden dimension sizes and layer number; (2) we increase the model size of ConfGF to 12.28M by increasing hidden dimension sizes, and re-run the experiments. The results are shown in Table 10.

| Dataset | QM9 | | | | Drugs | | | |
|---|---|---|---|---|---|---|---|---|
| | COV(%)↑ | | MAT (Å)↓ | | COV(%)↑ | | MAT (Å)↓ | |
| Methods | Mean | Median | Mean | Median | Mean | Median | Mean | Median |
| ConfGF (0.81M) | 88.49 | 94.13 | 0.2673 | 0.2685 | 62.15 | 70.93 | 1.1629. | 1.1596 |
| Ours (0.98M) | 80.47 | 95.83 | 0.2859 | 0.2508 | 79.24 | 90.91 | 1.0777 | 1.0665 |
| ConfGF (12.28M) | 86.86 | 93.49 | 0.3377 | 0.3450 | 55.36 | 58.20 | 1.2186 | 1.2134 |
| Ours (13.29M) | 81.44 | 96.61 | 0.2879 | 0.2428 | 77.78 | 86.09 | 1.0657 | 1.0563 |

Table 10: Comparison of our method and ConfGF with different model sizes

We can see that: (i) when we reduce our model size to 0.98M, our method still outperforms ConfGF on GEOM-Drugs; on GEOM-QM9, our method outperforms ConfGF evaluated by the median

values of COV and MAT. (ii) The performances of our model with 13.29M parameters and 0.98M parameters are similar. (iii) When we increase the model size of ConfGF, the performance becomes worse, which shows that ConfGF cannot benefit from more parameters. We observe that a larger ConfGF (12.28M) suffers from larger training loss than a smaller ConfGF (0.98M), which shows its limitation.

## C.5    MORE DISCUSSIONS ON THE CONFORMATION WITH MORE HEAVY ATOMS

In Table 3, we observe that our method works better than distance-based methods on molecules with more heavy atoms. Our conjecture is that for these distance-based works, they usually extend the molecular graph with 1,2,3-order neighbors, which is sufficient to determine the 3D structure in principle. For GEOM-QM9 dataset, considering the number of atoms is less than 10, this extended graph is nearly a complete graph and can provide enough signals to reconstruct the 3D structure. Therefore, these distance-based performances are good on GEOM-QM9 dataset. For GEOM-Drugs dataset, the numbers of atoms are much more than those in GEOM-QM9. Although in theory, the distances in a third-order extended graph can reconstruct the 3D structure, practically the signals are still not enough. Our method does not rely on the interatomic distances, and can achieve good results on large molecules.

