# OpenReview forum: "Direct Molecular Conformation Generation"
_ICLR.cc/2022/Conference — ICLR 2022 Submitted_

### Official Review · Reviewer_T1Ea · 2021-11-02

**Correctness:** 4
**Technical Novelty And Significance:** 3
**Empirical Novelty And Significance:** 2
**Recommendation:** 6
**Confidence:** 3

**Main Review:**

Strengths:
* The results are strong and the presented methodology is clear.
* The ablations shown in Table 5 are very nice to see and I think are a useful part of the paper.

Weaknesses:
* Some of the discussion of results could be expanded, or at least included in an appendix. For example, "This is consistent with our conjecture, and also demonstrates the potential of directly generating coordinates for complex molecules with more heavy atoms." This would be nice to have been further expanded upon.
* The presented results are limited in scope, and largely rely on comparisons with ConfGF, though then some additional comparisons would be useful. For example, for the property prediction showing results comparing to Tables 3 and 4 in the ConfGF paper would be useful.

Comments:
* In Fig. 3, what do the colors represent? Are two views shown?
*  A table with exact hyper parameters/etc in the appendix could be useful (in addition to the details in section 4.1). It's great that code is provided.
* Some methods, like G-SchNet (https://github.com/atomistic-machine-learning/G-SchNet) from my understanding do not produce confirmations which violate the triangle inequality. Is this a misunderstanding on my end, or is this approach of successively predicting distances an alternate approach that produces valid molecules?  I realise it is a different problem being approached, but molecules are still being generated.
* In Table 2, what is your interpretation of the COV(%) being 100%?
* For the property prediction, on QM9 various properties can be predicted. Could these all be shown?
* The text is at times a bit difficult to follow, but it does not detract from the ability to follow the work as a whole.
* I don't quite follow the "ours with /FF" in Table 5, and why this is not the "default" flavour?

**Summary Of The Paper:**

The authors present a method that is able to accurately generate molecule confirmations, achieving competitive results across four datasets.
Unlike previous methods that often rely on directly predicting various inter-atomic distances (through a distance matrix) (which in some cases results in physically impossible molecules), the presented work predicts the location of all atoms. The loss function is invariant to rotation, an important property of many molecules.

**Summary Of The Review:**

The work presents strong results on molecular confirmation prediction. A novel method is developed that mitigates concerns of many previous methods by ensuring adherence to geometric rules. The results shown are very competitive, though I would like to see slightly broader comparisons and more exposition on various points that are raised in discussion.

---

> ### Author Response · Authors · 2021-11-18
> **Response to Reviewer T1Ea**
>
> Thanks for your review comments!
> > [Q1] Some of the discussion of results could be expanded, or at least included in an appendix. For example, "This is consistent with our conjecture, and also demonstrates the potential of directly generating coordinates for complex molecules with more heavy atoms." This would be nice to have been further expanded upon.
>
> [R1] For these distance-based works, they usually extend the molecular graph with 1,2,3-order neighbors, which is sufficient to determine the 3D structure in principle. For GEOM-QM9 dataset, considering the number of atoms is less than 10, this extended graph is nearly a complete graph and can provide enough signals to reconstruct the 3D structure. Therefore, these distance-based performances are good on GEOM-QM9 dataset. For GEOM-Drugs dataset, the numbers of atoms are much more than those in GEOM-QM9. Although in theory, leveraging the distances in a third-order extended graph can reconstruct the 3D structure, practically the signals are still not enough. Our method does not rely on the interatomic distances and can achieve good results on large molecules. We put the discussion in Appendix C.5..
>
> > [Q2] The presented results are limited in scope, and largely rely on comparisons with ConfGF, though then some additional comparisons would be useful. For example, for the property prediction showing results comparing to Tables 3 and 4 in the ConfGF paper would be useful.
>
> [R2] Thanks for your suggestions. We also make tables like the Table 3 and Table 4 in the ConfGF paper. Please kindly check Table 8 and Table 9 of our paper.
>
> Specifically, we randomly choose 30 molecules from GEOM QM9 and Drugs test sets and calculate these properties in ConfGF paper. We have released our code on Github.
>
> The results are shown in Appendix C.2. The results are consistent with our conclusion in paper. In general, our method still outperforms the baselines.
>
>
> > [Q3] In Fig. 3, what do the colors represent? Are two views shown?
>
> [R3] Thanks for pointing out this problem. The colors are shown by PyMol randomly. Now we have fixed the color according to the atom type and updated the Fig.3.
>
> > [Q4] A table with exact hyper parameters/etc in the appendix could be useful (in addition to the details in section 4.1). It's great that code is provided.
>
> [R4] We have added this table in Appendix B (Table 6).
>
> > [Q5] Some methods, like G-SchNet (https://github.com/atomistic-machine-learning/G-SchNet) from my understanding do not produce confirmations which violate the triangle inequality. Is this a misunderstanding on my end, or is this approach of successively predicting distances an alternate approach that produces valid molecules? I realise it is a different problem being approached, but molecules are still being generated.
>
> [R5] Yes, this work predicts the position of each atom in an autoregressive way, and in this way, the point coordinates would not conflict. As you pointed, it is a different problem from ours: G-SchNet does not take the 2D molecular graph as input, but to generate a 3D point cloud with desired property.
>
> > [Q6] In Table 2, what is your interpretation of the COV(%) being 100%?
>
> [R6] According to the definition of COV,  Equation (11) in our paper, the median COV(\%) being 100\% means that for more than half of the reference molecules, these exist generated molecules that are close to them within a predefined threshold. We add the discussion to the paper.
>
> > [Q7] For the property prediction, on QM9 various properties can be predicted. Could these all be shown?
>
> [R7] Yes, we also update these properties for QM9 dataset in Appendix C.2. Generally, our method performs the best.
>
> > [Q8] I don't quite follow the "ours with /FF" in Table 5, and why this is not the "default" flavour?
>
> [R8] The “with FF” means that we will refine the generated molecules with Force Field method in RDKit, which is always used in previous work. We just follow ConfGF and did not use “with FF” as the default setting.

---

> > ### Comment · Reviewer_T1Ea · 2021-11-20
> > **Thanks for your reply.**
> >
> > While I still think there is room for improvement, I am raising my score from a 5 to a 6, or marginally above the acceptance threshold.

---

### Official Review · Reviewer_rZ2g · 2021-11-02

**Correctness:** 2
**Technical Novelty And Significance:** 2
**Empirical Novelty And Significance:** Not applicable
**Recommendation:** 5
**Confidence:** 5

**Main Review:**

**Originality:** The work suffers from the lack of originality. The described model shares the same idea as ConfVAE and the unnamed model, proposed in [1] and [2] respectively - the autoencoder-based framework for conformation generation, the iterative generation procedure, and RMSD objective were proposed earlier. The main novelty of the model is neural network architectures of encoder and decoder, that utilize advanced graph convolutions. Both of [1] and [2] works are missed in baselines, the model compared with.

**Clarity and Quality:** The authors clearly describe their motivation and provide good discussion and ablation studies section, still paper contains vague contributions part and missed related work section. The paper is written in a hard-to-read way. The paper contains plenty of inline equations, inconvenient formatting of enumerated statements that make it hard to navigate over text and understand the method. The paper focuses too much attention on technical details of the model, that are minor and can be moved to the Supplementary materials.

**Significance:** The novelty of the paper is minor. The experiment results show the increase of metric values between the proposed model and ConfGF baseline, still, it's hard to determine the significance of the performance gap due to the lack of standard deviation. Also, the significance of the results is doubtful since the important baselines [1] and [2] are missed in Experiment section.

**Drawbacks / questions**:

1. I would recommend using different Greek letters to denote the encoders/decoder parts instead of subscript text to increase the readability of formulas.
2. The ablation studies cover only architecture details. Still [2] provide loss function based on distance differences that is also roto-translation invariant but do not require additional subproblem solving.
3. The small-scale dataset results are redundant still large-scale dataset covers it. I would recommend training CVGAE and CGCF on the large-scale dataset and including the results of these models.
4. The choice of conformation colors in Figure 3 is misleading. I would recommend to add an additional figure with aligned conformation cloud to show the diversity of generated conformations.

[1] An End-to-End Framework for Molecular Conformation Generation via Bilevel Programming, Xu et al.

[2] Auto-Encoding Molecular Conformations, Winter et al.

**Summary Of The Paper:**

The authors propose the generative model that constructs a conformation (set of 3d coordinates) of the molecular graph with a variational autoencoder (VAE) framework. Unlike several recent works on conformation generation based on generating of distance matrix and further conformation recovering by solving a time-consuming distance geometry problem, the proposed approach directly generates 3d coordinates with an iterative process inside the decoder part of the model.

**Summary Of The Review:**

The idea of the paper is not novel. The experiment part misses several important baselines. The text should be revised to improve readability and clarity. The ablation study section should provide experiments to justify loss function choice.

---

> ### Author Response · Authors · 2021-11-18
> **Response to Reviewer rZ2g (Part 1)**
>
> Thanks for your review comments!
> > [Q1] About the originality in main review. “described model shares the same idea as ConfVAE and the unnamed model, proposed in [1] and [2] respectively - the autoencoder-based framework for conformation generation, the iterative generation procedure were proposed earlier.
>
> [R1] The claim is wrong from the following aspects:
> 1. Our method is to directly generate coordinates of atoms, while [1] and [2] still generate distances or angles first and then reconstruct the coordinates based on them. Specifically, Reference [1] (i.e., ConfVAE) refines the generated conformation based on the interatomic distances (See the algorithm 1 in Appendix B of Reference [1]), while our method directly generates coordinates without generating interatomic distances. Reference [2] is about to predict the bond lengths, bond angles and dihedral angles, which is quite different from ours. Therefore, from the motivation and implementation, our method is significantly different from theirs.
> 2. As far as we can find, [2] does not have an iterative generation procedure at all. In [1], the iterative refinement of coordinates is the inner loop of the optimization by running multiple gradient descent, but the  distances (or coordinates) are only output at the last layer of the model. In comparison, our decoder outputs the coordinates at each block, and the output of the last block is refined by the next layer.
>
> > [Q2] Comparison with reference [1] and [2]
>
> [R2] We run the baseline ConfVAE [1], and the results are shown in the table below. No matter for small-scale GEOM-QM9 and GEOM-Drugs and the large-scale version, our method significantly outperforms [1]. We also update the results in Table 1 and Table 2 of our paper.
>
> |Small Scale| QM9| | | |Drugs| | | |
> |---|---|---|---|---|---|---|---|---|
> ||Cov(%) $\uparrow$||MAT (\AA) $\downarrow$||Cov(%) $\uparrow$||MAT (\AA)$\downarrow$||
> |Metric|Mean|Median|Mean|Median|Mean|Median|Mean|Median|
> |ConfVAE|80.42|85.31|0.4066|0.3891|53.14|53.98|1.2392|1.2447|
> |Ours|81.44|96.61|0.2879|0.2428|     77.78 |  86.09 |  1.0657|  1.0563|
>
> |Large Scale| QM9| | | |Drugs| | | |
> |---|---|---|---|---|---|---|---|---|
> ||Cov(%) $\uparrow$||MAT (\AA) $\downarrow$||Cov(%) $\uparrow$||MAT (\AA)$\downarrow$||
> |Metric|Mean|Median|Mean|Median|Mean|Median|Mean|Median|
> |ConfVAE|80.18|85.87|0.3684|0.3776|57.63|63.75|1.2125|1.1986|
> |Ours|86.58|100.00|  0.2457 | 0.1756  |   90.68  |100.00  |0.8934 | 0.8703|
>
> Reference [2] does not provide open source code, which makes it hard for us to implement it. Reference [2] does not provide a table to show the results on GEOM-QM9 and GEOM-drugs, and does not compare with previous methods like CGCF, ConfGF, etc. Therefore, currently, we are not able to compare with reference [2].
>
> > [Q3]Clarity and Quality:
>
> [R3] Thanks for your suggestions, and we will rewrite these parts.
>
> > [Q4] it's hard to determine the significance of the performance gap due to the lack of standard deviation
>
> [R4] In previous paper, nobody reports standard deviation. Follow your suggestions, we have repeated our method for 5 times with different seeds, the results are shown below.
>
> |Dataset| QM9| | | |Drugs| | | |
> |---|---|---|---|---|---|---|---|---|
> ||Cov(%) $\uparrow$||MAT (\AA) $\downarrow$||Cov(%) $\uparrow$||MAT (\AA)$\downarrow$||
> |Metric|Mean|Median|Mean|Median|Mean|Median|Mean|Median|
> |ConfGF|88.49|94.13|0.2673|0.2685|62.15|70.93|1.1629|1.1596|
> |Ours|81.57 $\pm$ 0.18 |96.95 $\pm$ 0.20|0.2871 $\pm$ 0.0010|0.2442 $\pm$ 0.0021|77.76 $\pm$ 0.28|85.50 $\pm$ 0.37|1.0649 $\pm$ 0.0035|1.0487 $\pm$ 0.0043|
>
> We can see that the gaps between our method and ConfGF are much larger than the standard deviation.
>
> > [Q5]  Also, the significance of the results is doubtful since the important baselines [1] and [2] are missed in Experiment section.
>
> [R5] Please refer to [R2].

---

> > ### Author Response · Authors · 2021-11-18
> > **Response to Reviewer rZ2g (Part 2)**
> >
> > > [Q6] I would recommend using different Greek letters to denote the encoders/decoder parts instead of subscript text to increase the readability of formulas.
> >
> > [R6] Thanks for the suggestion. We will revise them to $\varphi$, $\phi$, $\psi$. For the readability of other reviewers, we will revise them after the review period.
> >
> > > [Q7] Still [2] provide loss function based on distance differences that is also roto-translation invariant but do not require additional subproblem solving.
> >
> > [R7] Reference [2] does not compare with any baseline, including CGCF, ConfGF. Reference [2] does not provide open sourced code. It is hard for us to re-implement it. Therefore, in our paper, we are not able to directly compare with it.
> >
> > However, following your comments, we try to add the loss functions in [2] as auxiliary loss functions of our method, and observe some improvement. The detailed results are in Appendix C.1. Note that, even with the loss function in [2], we still directly generate coordinates, not generating distances/angles first and then reconstructing conformations like [2].
> >
> > > [Q8] The small-scale dataset results are redundant still large-scale dataset covers it. I would recommend training CVGAE and CGCF on the large-scale dataset and including the results of these models.
> >
> > [R8] We have updated the results for CVGAE and CGCF on the large-scale dataset. Our method still significantly outperforms them. Please kindly check Table 2 in our paper.
> >
> > > [Q9] The choice of conformation colors in Figure 3 is misleading. I would recommend to add an additional figure with aligned conformation cloud to show the diversity of generated conformations.
> >
> > [R9] Thanks for your suggestion, we have fixed the color in Figure 3 according to the atom type, and for each conformation, we have aligned it with the first reference conformation. The three different conformations of each molecule in Figure 3 shown the diversity of each method.

---

> > > ### Comment · Reviewer_rZ2g · 2021-11-25
> > > **I thank the authors for the detailed comments.**
> > >
> > > The authors answered the majority of questions and updated the paper in correspondence to the provided feedback. Still, I stay with the opinion that the article lacks novelty. I will raise my score from 3 to 5.

---

> > > > ### Author Response · Authors · 2021-11-26
> > > > **Reply to the novelty of our work**
> > > >
> > > > Thanks for your reply. We would like to emphasize our novelty as follows:
> > > >
> > > > (1) Although there are many works for conformation generation, we are the first to demonstrate that directly generating coordinates without dependency on interatomic distance is feasible, and it is better than distance-based method empirically.
> > > >
> > > > (2) We propose a new attention-based graph model, which is demonstrated to be effective in conformation generation task.
> > > >
> > > > (3) Inspired by Reviewer P4XU, we improve our loss function with symmetric atoms permutation into consideration and design a new loss function at (https://openreview.net/forum?id=kcrIligNnl&noteId=Fy_Km31mrA). The performance of our new loss function is significantly better than prior distance-based method.
> > > >
> > > > |Dataset| QM9| | | |Drugs| | | |
> > > > |---|---|---|---|---|---|---|---|---|
> > > > ||Cov(%) $\uparrow$||MAT (\AA) $\downarrow$||Cov(%) $\uparrow$||MAT (\AA)$\downarrow$||
> > > > |Metric|Mean|Median|Mean|Median|Mean|Median|Mean|Median|
> > > > |ConfGF|88.49|94.13|0.2673|0.2685|62.15|70.93|1.1629|1.1596|
> > > > |Ours|81.44|96.61|0.2879|0.2428|77.78|86.09|1.0657|1.0563|
> > > > | Ours w/ permutation invariance  |95.53|99.42|0.2082|0.2069|96.28|100.00|0.7308|0.7295|

---

> ### Author Response · Authors · 2021-11-22
> **Thanks again, any further comments?**
>
> Dear Reviewer,
>
> Thanks for your review comments. Do you have further comments or any concerns to our paper?
>
> Regards.

---

### Official Review · Reviewer_P4XU · 2021-11-02

**Correctness:** 3
**Technical Novelty And Significance:** 2
**Empirical Novelty And Significance:** 2
**Recommendation:** 3
**Confidence:** 3

**Main Review:**


Overall, this work appears to be an advance compared to the
Langevin-based method it mainly compares against.  It is not clear to
me if this is simply due to a different parameter setting (e.g. a
larger model than a year ago works better for larger molecules), or if
this improvement is due to a genuine algorithmic advance, and if so,
what part of the algorithm helps the larger molecules the most and why
couldn't the authors beat the previous model on the smaller-sized
molecules.  I appreciated the authors effort to document the change of
the results across the methods in Table 3, but it would help if they
could also modify their new model (or ConfGf, or both) to a
drastically different size.

I did not understand what was the contribution for the fine-grained
loss function used in training. Unless I misunderstand something
important, the particular matching loss is not invariant to
permutations of symmetric atoms---think of the different ways to
enumerate the atoms in a methyl group, or replace the hydrogen in a
methane by methyls, or take any symmetric groups in a molecule that
can generate a combinatorial number of equivalent coordinates that
represent the same molecule graph. So I don't understand the statement
of the authors that this loss is zero if and only if R1 and R2
represent the same molecule structure---what would the loss be if I
randomly permute the symmetric hydrogen identities but not change the
coordinates?

Regarding Secion 4.3, I understand that these types of tables have
been published previously, but I wanted to state that I strongly
disagree with the terminology of calling the average energy of the
generated conformers a "property prediction".  I don't think that
there exists a single experiment that can measure this particular
quantity, although arguably one could use a Quantum Mechanical code,
as these authors and others have done, and calculate a number for it.
From a physics perspective, the somewhat more meaningful property is
the Boltzmann-weighted average of the energies of the molecules---is
that perhaps the quantity that this benchmark is supposed to evaluate?
Furthermore, what is the meaning of switching to an MMFF force field
for optimization and then use Psi4---why not directly evaluate the
conformation with Psi4?  Finally, what are the units in the energies
of Table 4?  If these are in Hartree (which I thought were the default
units for Psi4, though I'm no expert in it), then the reported errors
are all huge to be meaningful one way or another (to the order of more
than 13 kcal/mol for the smallest number listed in that table); these
types of enormous errors have little meaning for any conformational
energy comparison, as a trivial small bond-length deviation could add
this level of energy, whereas a wrong dihedral angle couldn't (unless
parts of the molecule almost clashed onto itself).  Perhaps the
authors would want to either clarify section 4.3 further (it took a
while to parse what the authors actually calculate there, and from a
quick check of the repo, I couldn't locate the relevant code for it)
or simply drop it altogether since it probably distracts from the rest
of this work.

I also didn't understand the first two of the three future directions:
I am not clear what the authors mean when they suggest combining this
method with flow-based or Langevin dynamics models---can they clarify
this point?  Importantly, what are the cases that fail in the current
model---can the authors document some examples in an appendix?



**Summary Of The Paper:**


The paper presents a direct method for generating a molecular
conformation conditional on the molecular graph.  Inspired by the
success of AlphaFold2, the authors demonstrated that one could
directly write out 3D coordinates for a small molecule rather than use
the previous ways of generating intermediate interatomic distance
matrices or iterate over forces. The method displays a decent
performance on the standard benchmark set of QM9.  It performs better
than past baseline methods for larger molecules.


**Summary Of The Review:**


I think that this paper is an advance on previous work and probably represents
the state of the art in generating molecular conformations conditional
on the graph.  The paper could use a review of the language to clarify
certain aspects of it, but I don't think that it is strong enough for ICLR.
Importantly, it should attempt to clarify if the improvement is mainly
due to model size changes or due to algorithm changes.

---

> ### Author Response · Authors · 2021-11-18
> **Response to Reviewer P4XU (Part 1)**
>
> Thanks for your review comments!
> > [Q1] It is not clear to me if this is simply due to a different parameter setting
>
> [R1] The improvement is not due to a different parameter setting. In the initial submission, we did not tune the model size in purpose. The model size of our model is 13.29M, which is indeed larger than the ConfGF model (0.81M). After that, (i) we reduce our model size to 0.98M by reducing hidden dimension sizes and layer number; (ii) we increase the model size of ConfGF to 12.28M by increasing hidden dimension sizes, and re-run the experiments. The results are shown as follows.
>
> |Dataset| QM9| | | |Drugs| | | |
> |---|---|---|---|---|---|---|---|---|
> ||Cov(%) $\uparrow$||MAT (\AA) $\downarrow$||Cov(%) $\uparrow$||MAT (\AA)$\downarrow$||
> |Metric|Mean|Median|Mean|Median|Mean|Median|Mean|Median|
> |ConfGF(0.81M)|88.49|94.13|0.2673|0.2685|62.15|70.93|1.1629.|1.1596|
> |Ours(0.98M)|80.47|95.83|0.2859|0.2508|79.24|90.91|1.0777|1.0665|
> |ConfGF(12.28M)|86.86|93.49|0.3377|0.3450|55.36|58.20|1.2186|1.2134|
> |Ours(13.29M)|81.44|96.61|0.2879|0.2428|77.78|86.09|1.0657|1.0563|
>
> We can see that: (i) when we reduce our model size to $0.98$M, our method still outperforms ConfGF on GEOM-Drugs; on GEOM-QM9, our method outperforms ConfGF evaluated by the median values of COV and MAT. (ii) The performances of our model with 13.29M parameters and 0.98M parameters are similar. (iii) When we increase the model size of ConfGF, the performance becomes worse, which shows that ConfGF cannot benefit from more parameters. We observe that a larger ConfGF (12.28M) suffers from larger training loss than a smaller ConfGF (0.98M), which shows its limitation.
> These results demonstrate that improvement is from our model design, which is one of our contributions, not using more parameters. These results have been added to Appendix C.4.
>
>
> > [Q2] what part of the algorithm helps the larger molecules the most and why couldn't the authors beat the previous model on the smaller-sized molecules.
>
> [R2] It is directly generating coordinates without using distances that beats previous baselines. For the distance-based works, they usually extend the molecular graph with 1,2,3-order neighbors, which is sufficient to determine the 3D structure in principle. For GEOM-QM9 dataset, considering the number of atoms is smaller than 10, this extended graph is nearly a complete graph and can provide enough signals to reconstruct the 3D structure. Therefore, these distance-based performances are good on GEOM-QM9 dataset. For GEOM-Drugs dataset, the numbers of atoms are much more than those in GEOM-QM9. Although in theory, leveraging the distances in a third-order extended graph can reconstruct the 3D structure, practically the signals are still not enough. Our method does not rely on the interatomic distances and can achieve good results on large molecules.
>
> Currently, for the QM9 results of Table 1 and Table 2, we can see that in most cases (5 / 8), our method still beats baselines rather than “why couldn't the authors beat the previous model on the smaller-sized molecules”. For the three cases that our method is not as good as ConfGF, our conjecture is that ConfGF leverages more information (i.e.. the distances up-to-third order neighbors) to reconstruct the 3D structure. We also add another two constraints to the output conformations: one is that bond length should be close to the groundtruth, and the other is about bond angle. After applying those two auxiliary losses, our method significantly outperforms all distance-based methods. The details are in Appendix C.1., please kindly have a check.

---

> > ### Author Response · Authors · 2021-11-18
> > **Response to Reviewer P4XU (Part 2)**
> >
> > >[Q3] I did not understand what was the contribution for the fine-grained loss function used in training. Unless I misunderstand something important, the particular matching loss is not invariant to permutations of symmetric atoms---think of the different ways to enumerate the atoms in a methyl group, or replace the hydrogen in a methane by methyls, or take any symmetric groups in a molecule that can generate a combinatorial number of equivalent coordinates that represent the same molecule graph. So I don't understand the statement of the authors that this loss is zero if and only if R1 and R2 represent the same molecule structure---what would the loss be if I randomly permute the symmetric hydrogen identities but not change the coordinates?
> >
> > [R3] We have several claims towards this question.
> > 1. We indeed know this case. Therefore, in our implementation, we only predict the positions of heavy atoms and do not predict the positions of hydrogen atoms. This is consistent with evaluation metrics used in previous works. Note that Alphafold2 does not generate hydrogen atoms neither. Without hydrogen atoms, the problem of permutations invariance can be greatly simplified.
> > 2. The purpose for we propose the matching loss, is to make our loss invariant to global transformation of one rigid conformation. To solve the problem of random permutation of any symmetric atoms , we can follow AlphaFold2. Specifically, we enumerate all the permutations of these symmetric atoms or chiral atoms, calculate their loss to the ground-truth conformation, and choose the smallest one as the final loss. However, as the first work that successfully generating coordinates directly and outperforming distance-based method, we can regard more fine-grained loss function as future work.
> > 3. “So I don't understand the statement of the authors that this loss is zero if and only if R1 and R2 represent the same molecule structure” => this statement is not perfect. In the new version, we change it to “… if R1 is obtained by a roto-translational operation of R2”.
> >
> > > [Q4] Regarding Section 4.3, I understand that these types of tables have been published previously, but I wanted to state that I strongly disagree with the terminology of calling the average energy of the generated conformers a "property prediction". From a physics perspective, the somewhat more meaningful property is the Boltzmann-weighted average of the energies of the molecules---is that perhaps the quantity that this benchmark is supposed to evaluate
> >
> > [R4] Just as you mentioned, we follow prior work to name Section 4.3 and calculate numbers. Moreover, from a machine learning perspective, if the training data is sampled from Boltzmann distribution, then the average energy of our randomly sampled data is the approximation of the Boltzmann-weighted average energy.
> >
> > >[Q5] Furthermore, what is the meaning of switching to an MMFF force field for optimization and then use Psi4---why not directly evaluate the conformation with Psi4?
> >
> > [R5] In Table 4, the baseline “Rdkit” is with MMFF optimization, and we did this optimization for each method by default. Per your request, we also conduct this experiment without MMFF optimization, and the results are shown in Table 10. As the table shows, our method also achieves the best performance in general.

---

> > > ### Author Response · Authors · 2021-11-18
> > > **Response to Reviewer P4XU (Part 3)**
> > >
> > > > [Q6] Finally, what are the units in the energies of Table 4? If these are in Hartree (which I thought were the default units for Psi4, though I'm no expert in it), then the reported errors are all huge to be meaningful one way or another (to the order of more than 13 kcal/mol for the smallest number listed in that table); these types of enormous errors have little meaning for any conformational energy comparison, as a trivial small bond-length deviation could add this level of energy, whereas a wrong dihedral angle couldn't (unless parts of the molecule almost clashed onto itself). Perhaps the authors would want to either clarify section 4.3 further (it took a while to parse what the authors actually calculate there, and from a quick check of the repo, I couldn't locate the relevant code for it) or simply drop it altogether since it probably distracts from the rest of this work.
> > >
> > > [R6] The unit in Table 4 is eV, not Hartree. Please refer to https://github.com/DirectMolecularConfGen/DMCG/blob/main/confgen/utils/psi4_utils.py#L24
> > >
> > > We have updated the relevant code on Github. Please kindly refer to the commit ` update code for psi4`, whose url is
> > > https://github.com/DirectMolecularConfGen/DMCG/commit/1548d106b77eb127ed96674a06cc61a23322a943
> > >
> > > > [Q7] I also didn't understand the first two of the three future directions: I am not clear what the authors mean when they suggest combining this method with flow-based or Langevin dynamics models---can they clarify this point? Importantly, what are the cases that fail in the current model---can the authors document some examples in an appendix?
> > >
> > > [R7]
> > > 1. The generative model in our work is variational auto-encoder (VAE). The flow-based generative model is also widely used that explicitly models a probability distribution by leveraging normalizing flow. To use flow-based model, we just need to replace workflow in Section 3.2 to the flow-based. The model in Section 3.3 and the loss function in Eqn.(1) remain unchanged. To use Langevin dynamics, a solution is to follow (Shi et al. 2021): (i) the scorenet outputs the log-density gradient of coordinates; (ii) the model backbone of (Shi et al. 2021) is replaced with the model we designed in Section 3.3; (iii) the coordinate update rule is the 8th step in Algorithm 1 of (Shi et al. 2021).
> > > 2. We have shown Failure case in Appendix C.3 and propose a preliminary solution for this failure case. Two examples are shown in Figure 5 of the paper. We can see that there exists a rotatable ring at the end of a molecule, where the ring is symmetric to the bond connecting itself to the rest of the molecule. Our method fails to generate the coordinates of such rings, but simply puts them in a line. This is because the model is trapped into local optimal. By using the additional loss functions as we introduced Appendix C.1, we can successfully recover the conformations of those rings (see the row Ours + Aux). We will keep exploring along this direction.

---

> > > ### Comment · Reviewer_P4XU · 2021-11-18
> > > **minor clarifications**
> > >
> > > Thanks for the additional clarifications; I will take a detailed look tomorrow night and check if you have an updated version of the manuscript by then.  I have a couple of quick questions before then:
> > >
> > > 1) unless I am very mistaken, I understand that your input data is *not* weighed from the Boltzmann distribution, otherwise, any room temperature sample with 0.1 eV higher than the ground state would appear about 100 times less frequently than the ground state (and, similarly, a sample with 0.2 eV higher, would appear 10,000 times less frequently than the ground state of your dataset)---is that not the case here, i.e. do you actually use Boltzmann weighted input samples in your method?  More importantly, I still don't understand what is the purpose of using MMFF optimization and then calculating energies?  If one wanted to optimize with MMFF, I'm sure there are ways to sample all possible minima and add the proper Botzmann weights to those samples (or not.)
> > >
> > > 2) AlphaFold2 doesn't have a hard problem with the permutations of symmetric atoms in the side chains, since the chemistry of amino acids is significantly simpler than the chemistry of arbitrary organic molecules and the problem can be decomposed into a set of independent local optimization problems without too much loss in accuracy.  However, your case is different.  Even if you excluded the hydrogen atoms (which was not clear from the previous version of the manuscript) the organic molecules often have very high permutation symmetries because of the combinatorics of different side chains in those molecules, or the symmetric branching of parts of the molecule.  Unless there is an unintentional symmetry-breaking data leakage during training, which I don't suspect you have, how can the model learn to make coordinates for any molecule with heavy atom symmetries (the majority of drug-like molecules) if the coordinates for an atom could be matching locations at very different final positions?   Is this lack of a proper loss function the reason why the generation fails to make a single ring in your answer to R7.2 below?
> > > (Also, as a minor note, and without having seen the paper, I don't think that your suggested change to the language is clear enough for a reader who hasn't seen this discussion and it avoids the main point of concern in your methodology.)

---

> > > > ### Author Response · Authors · 2021-11-19
> > > > **Response to Reviewer P4XU**
> > > >
> > > > Thanks for your quick reply!
> > > > > [Q1] Unless I am very mistaken, I understand that your input data is not weighed from the Boltzmann distribution.
> > > >
> > > > [R1] You are right. We just follow ConfGF to obtain the data and conduct experiments. In the current machine learning paper including ConfGF, CFCG, etc, none of them use the Boltzmann weight. The reason is that we are not able to obtain such weights. Can you give us a script to obtain the Boltzmann weight?
> > > >
> > > > > [Q2] More importantly, I still don't understand what is the purpose of using MMFF optimization and then calculating energies.
> > > >
> > > > [R2] Now, in Section 4.3, we remove MMFF, and re-run the experiments. Our method is still the best.
> > > >
> > > > > [Q3] About alphafold2
> > > >
> > > > [R3] Thanks for your comments on Alphafold 2. The reason we mentioned alphafold 2 is that, we can leverage the idea in Alphafold 2 to solve the permutation invariance problem. The current loss function cannot handle the permutation invariance. However, considering its superior performance than all previous distance-based method, we think it is enough as the first step and we will keep follow this direction.
> > > >
> > > > We are working on efficient GPU implementation about permutation invariant loss function, following [ref1].
> > > >
> > > > For the failure cases, it can be solved by using additional loss function, as we introduced in Appendix C.1.
> > > >
> > > > [ref1]  spyrmsd: symmetry-corrected RMSD calculations in Python, https://jcheminf.biomedcentral.com/track/pdf/10.1186/s13321-020-00455-2.pdf

---

> ### Author Response · Authors · 2021-11-24
> **Reply to the permutations of symmetric atoms problem**
>
> Thanks to your valuable review comments again!
>
> (1) Inspired from your comments, we realize that the permutation of symmetric atoms is indeed a problem for not only our method, but also the distance-based methods. Specifically, the distance-based methods need to calculate distances to the $2^{nd}$- and $3^{rd}$-order neighbors, which are not always the same for these symmetric atoms. Therefore, how to solve this permutation problem is important for the conformation generation task.
>
> (2) We solve this problem following our previous proposal in [R3] (https://openreview.net/forum?id=kcrIligNnl&noteId=lyMwB62ZlYe). We view all the graphs resulted from different permutation of symmetric atoms as the isomorphic graphs. From the graph theory perspective, an isomorphism between graph $A$ and graph $B$ is a bijective mapping $\sigma$, which maps the vertices of graph $A$  to vertices of graph $B$  while preserving the edge structures of the graphs. This is the same problem as ours. We compute all the losses of these isomorphic graphs, and choose the minimum one as the final loss to optimize our model. In this way, we take the permutation of symmetric loss functions into consideration. Specifically, the steps are shown as follows:
>
>     a. Given a molecular graph, we generate all the isomorphic graphs, using “graph-tool” (https://github.com/antmd/graph-tool).
>     b. We use our model to generate a conformation of the input molecular graph.
>     c. We permute the coordinates according to these isomorphic graphs, and calculate the loss for each one permutation.
>     d. We choose the minimum loss as the final loss, and back-propagate the gradient to update our model.
>
> The revised loss of our model becomes $\ell_M(R_1,R_2)=\min_{\rho, \sigma} \Vert\rho(\sigma(R_1))-R_2\Vert_F^2$, where $\sigma$ is the permutation of the corresponding isomorphic graph, and $\rho$ is the roto-translational operator. The detailed implementation is at https://github.com/DirectMolecularConfGen/DMCG/commit/6c8f3c6d2c6059400e0d7cc1ca3749f6e05d2bce
>
> The mean and median numbers of isomorphic graphs are 1.405 and 1 on GEOM-QM9 and 5.888 and 1 on GEOM-Drugs dataset, which requires acceptably additional training time. The results are shown below (denoted as "Ours w/ permutation invariance").
>
> |Dataset| QM9| | | |Drugs| | | |
> |---|---|---|---|---|---|---|---|---|
> ||Cov(%) $\uparrow$||MAT (\AA) $\downarrow$||Cov(%) $\uparrow$||MAT (\AA)$\downarrow$||
> |Metric|Mean|Median|Mean|Median|Mean|Median|Mean|Median|
> |ConfGF|88.49|94.13|0.2673|0.2685|62.15|70.93|1.1629|1.1596|
> |Ours|81.44|96.61|0.2879|0.2428|77.78|86.09|1.0657|1.0563|
> |Ours+Aux|91.97|98.66|0.2388|0.2231|92.45|98.70|0.8983|0.9016|
> |Ours w/ permutation invariance |95.53|99.42|0.2082|0.2069|95.74|100.00|0.7730|0.7719|
>
> By using the new loss function with permutation into consideration, the results are significantly better than all previous baselines. Please note that experiments on GEOM-Drugs dataset have not converged. We will update the final results once we get them.

---

> > ### Author Response · Authors · 2021-11-26
> > **Updates on our new loss function**
> >
> > Our experiments on GEOM-Drugs dataset are done. The final results are shown as follows,
> >
> > |Dataset| QM9| | | |Drugs| | | |
> > |---|---|---|---|---|---|---|---|---|
> > ||Cov(%) $\uparrow$||MAT (\AA) $\downarrow$||Cov(%) $\uparrow$||MAT (\AA)$\downarrow$||
> > |Metric|Mean|Median|Mean|Median|Mean|Median|Mean|Median|
> > |ConfGF|88.49|94.13|0.2673|0.2685|62.15|70.93|1.1629|1.1596|
> > |Ours|81.44|96.61|0.2879|0.2428|77.78|86.09|1.0657|1.0563|
> > |Ours+Aux|91.97|98.66|0.2388|0.2231|92.45|98.70|0.8983|0.9016|
> > | Ours w/ permutation invariance  |95.53|99.42|0.2082|0.2069|96.28|100.00|0.7308|0.7295|
> >
> > As the results shown, (1) With the symmetry permutation invariant loss function, our method is significantly better than prior state-of-the-art method, ConfGF, on GEOM-QM9 and GEOM-Drugs dataset. (2) On GEOM-Drugs dataset, where there are more isomorphic graphs, the influence of the symmetry permutation invariant loss function is more significant.

---

### Official Review · Reviewer_Y6xJ · 2021-11-03

**Correctness:** 3
**Technical Novelty And Significance:** 3
**Empirical Novelty And Significance:** 3
**Recommendation:** 6
**Confidence:** 5

**Main Review:**

Strengths:
The paper is well-organized and clearly written. All content is easy to understand for me.
The experimental part is comprehensive. The author nicely compares the proposed method with competitive baselines on both small-scale and large-scale datasets. The results demonstrate that the model is better or comparable with state-of-the-art results with much less computational cost.
Weaknesses:
The main concern of this paper is the originality. The main contribution of this paper is the decoder part, which is very similar to the “structure module” in Alphafold that iteratively updates the folding and node/edge features. However, I appreciate the efforts to extend the progress of protein folding to molecular modeling, so I also agree the adaption is non-trivial.
A minor problem: the problem definition in Sec.2 seems not accurate enough. In your statement, it seems each graph only corresponds to a single conformation, but this is not true. Please restate it.
I’m also interested in the design of the decoder block. I understand the training objective is invariant, however, I’m curious whether updating the coordinates with equivariant networks will further help the performance. Several recent works such as ConfGF have shown that equivariance is an important inductive bias for 3D generative models.


**Summary Of The Paper:**

The paper proposed a new VAE-based generative model for generating molecular conformations from graphs. The whole formulation and training objective are similar to existing VAE-based models like CVGAE, and the main contrition lies in the architecture of the decoder. The architecture borrows the idea from the recent advance of AlphaFold: the proposed decoder is composed of several identical updating blocks, where they iteratively update the node/edge embeddings as well as the coordinates in each block. And the blocks are implemented with advanced graph neural networks.


**Summary Of The Review:**

The paper proposed a VAE model for molecular conformation generation, where the decoder borrows the idea of Alphafold to iteratively refine the generated structure. Personally, the paper seems an extension of the recent progress of protein folding to the small molecule setting, but I appreciate the authors’ efforts to make it work. So currently I vote for a weak acceptance.

---

> ### Author Response · Authors · 2021-11-18
> **Response to Reviewer Y6xJ**
>
> Thanks for your review comments!
> > [Q1] A minor problem: the problem definition in Sec.2 seems not accurate enough. In your statement, it seems each graph only corresponds to a single conformation, but this is not true. Please restate it.
>
>
> [R1] Following the common practice (Shi et al. 2021), we define molecular conformation as it in our submitted version. According to your suggestions, we add a footnote to make it clear: A molecule can correspond to different conformations at different energy level. To model such cases, the problem is to generate $R$ based on $G$ and a random noise $z$.
>
> > [Q2] I’m also interested in the design of the decoder block. I understand the training objective is invariant, however, I’m curious whether updating the coordinates with equivariant networks will further help the performance. Several recent works such as ConfGF have shown that equivariance is an important inductive bias for 3D generative models.
>
> [R2] (1) In our preliminary experiments, it is hard to adapt the equivariant networks into our setting and we have not observed significant benefit of these equivariant networks. We will keep following this direction. (2) Inspired by your suggestions, we add a distance-based loss function and an angle-based loss functions, which are invariant to the roto-translation of coordinates. After using them, the performances are improved, as shown in the Table below (“Our + Aux” is the corresponding result). Please kindly refer to Appendix C.1 for more details.
>
> |Dataset| QM9| | | |Drugs| | | |
> |---|---|---|---|---|---|---|---|---|
> ||Cov(%) $\uparrow$||MAT (\AA) $\downarrow$||Cov(%) $\uparrow$||MAT (\AA)$\downarrow$||
> |Metric|Mean|Median|Mean|Median|Mean|Median|Mean|Median|
> |ConfGF|88.49|94.13|0.2673|0.2685|62.15|70.93|1.1629|1.1596|
> |Ours|81.44|96.61|0.2879|0.2428|77.78|86.09|1.0657|1.0563|
> |Ours+Aux|91.97|98.66|0.2388|0.2231|92.45|98.70|0.8983|0.9016|

---

### Decision · Program_Chairs · 2022-01-20

**Decision:**

Reject

**Comment:**

The paper proposed a new VAE-based generative model for generating molecular conformations from graphs. Reading the paper, the reviews and the rebuttal, it looks like this project is a work in progress and not yet ready for publication. Some reviewers indicate that the paper lacks significant technical novelty.

This is a worthy application. I encourage the authors to keep working and resubmit to another venue when they feel the work is ready and they have addressed all points raised by the reviewers.